# Clinical, Histological, and Molecular Features of Solitary Fibrous Tumor of Bone: A Single Institution Retrospective Review

**DOI:** 10.3390/cancers13102470

**Published:** 2021-05-19

**Authors:** Giuseppe Bianchi, Debora Lana, Marco Gambarotti, Cristina Ferrari, Marta Sbaraglia, Elena Pedrini, Laura Pazzaglia, Luca Sangiorgi, Isabella Bartolotti, Angelo Paolo Dei Tos, Katia Scotlandi, Alberto Righi

**Affiliations:** 1Department of Orthopedic Oncology, IRCCS Istituto Ortopedico Rizzoli, 40136 Bologna, Italy; giuseppe.bianchi@ior.it (G.B.); debora.lana88@gmail.com (D.L.); 2Department of Pathology, IRCCS Istituto Ortopedico Rizzoli, 40136 Bologna, Italy; marco.gambarotti@ior.it; 3Experimental Oncology Laboratory, IRCCS Istituto Ortopedico Rizzoli, 40136 Bologna, Italy; cristina.ferrari@ior.it (C.F.); laura.pazzaglia@ior.it (L.P.); katia.scotlandi@ior.it (K.S.); 4Department of Pathology, Azienda Ospedaliera di Padova, 35121 Padua, Italy; marta.sbaraglia@aopd.veneto.it; 5Department of Rare Skeletal Disorders, IRCCS Istituto Ortopedico Rizzoli, 40136 Bologna, Italy; elena.pedrini@ior.it (E.P.); luca.sangiorgi@ior.it (L.S.); isabella.bartolotti@ior.it (I.B.); 6Department of Medicine, University of Padua School of Medicine, 35121 Padua, Italy; angelo.deitos@unipd.it

**Keywords:** solitary fibrous tumor, primary bone tumor, risk stratification, prognosis, *NAB2-STAT6* fusion transcripts

## Abstract

**Simple Summary:**

Solitary fibrous tumors arising from the bone are an extremely rare event and only few cases have been previously described in the literature. It is characterized by a prominent, branched vascularization, with a thin and dilated vascular texture defined as “staghorn” and by the presence of the *NAB2-STAT6* gene rearrangement, present in about 90% of cases and considered a pathognomonic feature. In the present study, we described our series of 24 cases of primary solitary fibrous tumor of the bone to find any clinical and molecular prognostic factors and to compare them with those currently used for soft tissue solitary fibrous tumor and to evaluate the risk stratification system proposed by Demicco, in order to understand whether this system was able to correctly predict the risk of local and distant metastatic relapse even in the case of solitary fibrous tumor of the bone.

**Abstract:**

Primary solitary fibrous tumor (SFT) of the bone is extremely rare, with only few cases reported in the literature. We retrieved all cases of primary SFT of the bone treated at our institution and we assessed the morphology and the immunohistochemical and molecular features to investigate the clinical outcome of primary SFT of the bone and any clinical relevance of clinical and histological criteria of aggressiveness currently adopted for the soft tissues counterpart. Morphologically, 15 cases evidenced high cellularity, cytologic atypia, and foci of necrosis and were associated with more than 4 mitotic figures/10 HPF. Immunohistochemical analysis showed an expression of CD34 and of STAT6 immunopositivity in 95% and in 100% of cases, respectively. The presence of *NAB2-STAT6* chimeric transcripts was found in 10 out of 12 cases in which RT-PCR analysis was feasible, whereas *TERT* promoter mutations analysis was feasible in 16 cases and only a C-to-T substitution in a heterozygous state was found in one DNA sample for the C228T genetic variant. *P53* variants were assessed in 12 cases: 11 (91.6%) cases showed a variation, while in one case, no alteration was found. Disease-specific survival was 64% at 5 years and 49% at 10 years. Statistical analysis showed no correlation between survival and all the clinicopathological and molecular parameters evaluated. In conclusion, at difference to SFT of soft tissues, aggressive behavior of primary SFT of the bone seems to be independent from mitotic count or any other clinicopathological and molecular features.

## 1. Introduction

Solitary fibrous tumor (SFT) is a rare mesenchymal tumor of fibroblastic origin that can occur at any anatomic site and typically affects middle-aged adults [1,2,3]. It is characterized by a strong morphologic heterogeneity with a wide spectrum of biologic features. The histological and molecular diagnostic criteria used in soft tissue SFT (S-SFT) have been recently applied on “non otherwise classified” primary bone tumors, drawing out a new category of SFT of the bone (B-SFT) [4,5,6,7]. Nevertheless, B-SFT is exceedingly rare, with only few cases are described in the literature [8,9,10,11], and its biological behavior has not yet been assessed. From a histopathological and molecular point of view, primary B-SFT shares the same features of S-SFT. It is characterized by a prominent, branched vascularization, with a thin and dilated vascular texture defined as “staghorn” and by the presence of the *NAB2-STAT6* gene rearrangement (NGFI-A binding protein 2—Signal Transducer and Activator of Transcription 6), present in about 90% of cases and considered a pathognomonic feature [1,12]. Positivity to CD34 stain is distinctive in 90–95% of the cases. S-SFT has an intermediate malignant potential with a low risk of metastasis. Some studies have investigated the prognostic role of previously described molecular markers, without, however, obtaining conclusive results; the aforementioned prognostic criteria have never been explored in B-SFT [13,14,15,16]. Most S-SFTs are clinically indolent, with an intermediate malignant potential and a low risk of metastasis, showing an overall 5- and 10-year distant metastasis (DM)-free rates of 74% and 55%, respectively. In recent times, different stratification risk models have been proposed [17,18,19,20,21,22]. The current most utilized scoring system to discriminate different risk groups for S-SFT—also related to the development of distal metastasis—is the one proposed by Demicco et al. [22], which considers patient age, mitotic activity, tumor necrosis, and size. To date, few prognostic molecular markers have been analyzed. *NAB2–STAT6* chimeric transcripts, with a frequency ranging from 55 to 100% [23,24], and characterized by different breakpoints in fusion genes, might contribute to the morphologic diversity of SFT; some studies evidenced associations between specific fusion variants and different clinical features [21,25]. In addition, specific point mutations within the promoter region of telomerase reverse transcriptase (*TERT*)—C228Tand C250T—have been recently reported in S-SFT subsets and other tumors [15,25,26,27]. These mutations confer enhanced *TERT* promoter activity and have been suggested as predictive factors to identify high-risk patients. Finally, TP53 has also been proposed as an SFTs risk factor. In particular, tumors with TP53 mutations were almost always classified as high risk [21,28]. Due to the rarity of B-SFT and taking advantage of the availability of a large and homogeneous cohort of patients, the goal of this study was to better characterize the biological behavior of this specific SFT subset located in the bone considering both the clinical, histological, and molecular features, as well as the applicability of the risk stratification model used for S-SFT.

## 2. Materials and Methods

The study was carried out on 24 patients affected by primary B-SFT treated at the Istituto Ortopedico Rizzoli between 1970 and 2019.

All patients were investigated, excluding history of meningeal SFT, whose metastatic bone localization could be misdiagnosed with a primary B-SFT. All cases were retrieved both from a radiological and clinical point of view through a review of medical records (anatomical site, tumor size, type of treatment, and surgical margins in the operated patients) and defined with regard to both the immuno-histochemical profile (positivity for CD34 and/or STAT6) and the molecular one (presence of the *NAB2-STAT6* gene fusion products, of the C228T and C250T *TERT* promoter variants, and of mutations in the p53 gene). The Demicco model [22] was used for the patient’s risk stratification and the tumor size was assessed using the largest tumor dimension as a reference. All procedures were performed in accordance with the ethical standards of the Helsinki declaration. The study was approved by the ethical institutional committee on 22 July 2020 (study code: AVEC 730/2020/Oss/IOR). All analyses were completed with the help of the Statistical Package for Social Science (IBM Corp. Released 2013. IBM SPSS Statistics for Windows, Version 22.0. Armonk, NY, USA: IBM Corp.).

### 2.1. Histopathology Evaluation

Hematoxylin–eosin slides of all cases were reviewed by four pathologists (A.R., M.G., M.S., A.D.T.), and the morphological diagnosis of SFT was confirmed. Tumors were scored for mitotic figures, cellularity, nuclear pleomorphism, and presence of necrosis as universal and standardized criteria defining malignancy [17,18,27]. Mitotic index was calculated per 10 high-power fields (HPFs). The presence of high cellularity areas, defined as a hypercellular tumor with areas of nuclear overlap, and the presence of high pleomorphism, determined by hyperchromatic nuclei with foci of marked pleomorphism and bizarre cells according to Demicco criteria [22], were evaluated. Necrosis was scored as absence or minimal (<10%) or positive (≥10%), based on available histological sections.

### 2.2. Immunohistochemistry

All paraffin-embedded tumor samples were evaluated by immunohistochemistry, as previously reported [27], with the following antibodies: CD34 (QBEnd-10; Ventana Medical Systems) and STAT6 (S-20, SC-621; Santa Cruz Biotechnology, Inc., Dallas, TX, USA). First, 4-μm-thick tissue sections were cut, heated at 58 °C for 2 h, deparaffinized, and immunostained on a Ventana BenchMark following the manufacturer’s guidelines (Ventana Medical Systems, Tucson, AZ, USA). The detection was performed using the UltraView Universal Alkaline Phosphatase Red Detection Kit and the UltraView Universal DAB Detection Kit (Ventana Medical Systems).

### 2.3. DNA and RNA Isolation

Sixteen tumor samples were available for molecular analyses. DNA and RNA were isolated from 10 formalin-fixed paraffin-embedded (FFPE) and 6 frozen tissues by using a QIAamp DNA FFPE Tissue Kit (Qiagen, Valencia, CA, USA) and RNeasy FFPE Tissue Kit (Qiagen), DNAzol and TRIzol (Invitrogen, Carlsbad, CA, USA) in accordance with the manufacturer’s instructions.

### 2.4. Detection of NAB2-STAT6 Fusion Variants

The 24 most frequent *NAB2–STAT6* fusion variants found in S-SFT [27,29,30] were analyzed. PCR was performed by an AmplTaq Gold 360 Master Mix (Invitrogen) using 2 μL of cDNA product as previously described [27]. PCR products were sequenced using the BigDye Terminator v3.1 Cycle Sequencing Kit (Applied Biosystems) on an automated sequencer (ABI PRISM 3100 Genetic Analyzer 3130xl, Applied Biosystems, Foster City, CA, USA). To confirm the presence of specific *NAB2–STAT6* fusion breakpoints, the sequences were aligned using the CodonCode Aligner software (https://www.codoncode.com/aligner/, accessed on 29 April 2021).

### 2.5. TERT Promoter Mutation Analysis

The presence of C228T and C250T mutations at the *TERT* promoter region was primarily evaluated by Sanger sequencing as previously described [27].

Due to the low sensitivity of Sanger sequencing in detecting somatic mutations, we analyzed the same samples by Digital PCR (QuantStudio 3D Digital PCR System, Thermo Fisher Scientific, Waltham, MA, USA), used for rare allele detection to exclude the presence of *TERT* variants at low frequencies. We selected two TaqMan^®^ probe-based assays (Hs000000093_rm, Hs000000092_rm). Polymerase chain reaction amplification was carried out on a ProFlex™ 2 × flat PCR System (Thermo Fisher Scientific, Waltham, MA, USA). Subsequent analysis and post-processing were performed by the QuantStudio™ 3D AnalysisSuite™.

### 2.6. Analysis of p53 Mutation

To evaluate the presence of TP53 mutations, the samples were genotyped by direct sequence of all coding exons (2–11), including flanking intron-exon junctions. Sanger sequencing was performed using the BigDye Terminator v3.1 Cycle Sequencing Kit (Thermo Fisher Scientific) and the ABI PRISM 3500xL Geentic Analyzer (Thermo Fisher Scientific). To evaluate the presence of potential somatic copy number variations, a Microfluidic Chip-Based Digital PCR reaction was performed using the QuantStudio™3D Digital PCR System (QS3D, Thermo Fisher Scientific—US). A Taqman copy number assay was selected (Hs06423639_cn) to cover approximately the central part of the gene within exon 4 (location: hg38, Chr.17:7668402-7687550). RNase P gene was chosen as a reference locus. Polymerase chain reaction amplification was carried out on a ProFlex ™ 2 × Flat PCR System (Applied Biosystems). The fluorescence data were read and analyzed using QuantStudio 3D Analysis Suite Cloud Software. Results are expressed as copies per microliter and compared as a ratio of target (FAM)/Total (FAM + VIC) expressed in percentage. In case of a regular biallelic status, we expect this value to be around 50%. The TaqMan Copy Number probe was previously tested and validated on 6 DNA with a regular biallelic status of the p53 gene (with a target/total percentage range of 48.484–50.357%), confirming the absence of CNV alterations.

### 2.7. Statistics

Correlations between clinical, pathological, immuno-histochemical, and molecular data were assessed using contingency tables and chi-square test. The Kaplan–Meier method was used to estimate disease-specific survival (DSS), recurrence-free survival (RFS), and metastasis-free survival (MFS) based on histopathological criteria and the presence of *NAB2/STAT6* fusion variants.

MFS and DSS intervals were defined as the time between surgery and the first metastasis and death, respectively, or last follow-up available. Patients who died of other causes were excluded. By the log-rank test, differences in survival rates were assessed, considering *p* values < 0.05 as significant. For all analysis, was used the Statistical Package for Social Science (IBM Corp. Released 2013. IBM SPSS Statistics for Windows, Version 22.0. Armonk, NY, USA: IBM Corp.).

## 3. Results

### 3.1. Clinicopathological Evaluation

The clinical and pathological features of the 24 patients included in this study are summarized in Table 1 and Table 2. The cohort is composed of 14 females (58.3%) and 10 males (41.7%) ranging from 7 to 84 years (mean 51 years). Most tumors arose in the axial skeleton (4 sacrum, 4 pubis, 2 scapula, and one lumbar vertebra), 9 in the lower extremities (6 femur, 2 tibia, and one fibula), and 4 in the upper extremities (humerus). In 5 out of 24 cases who were not feasible for surgery, only a biopsy was performed, followed by radiation therapy in two cases, chemotherapy in one, association of chemo- and radiation therapy in one, and embolization in one case. Nineteen patients underwent segmental resection or amputation with wide/radical margins in 16, intralesional margins in 2, and with marginal margins in one patient.

In the group of 19 patients surgically treated, 3 patients (16%) developed local recurrence at a mean time of 106 months (range 72–149 months, median 97 months).

Twelve patients (50%) had metastasis (9 localized at lungs and 3 to bone): 3 patients had metastasis at presentation (in one case, lung; in one case, soft tissues; and in one case, both lungs and bone): chemotherapy (CT) with a combination of doxorubicin, methotrexate, cisplatin, and ifosfamide was given to the two patients with lung metastasis at presentation. The other 9 (37.5%) patients developed metastasis at a mean time of 53 months (range 3–108 months).

Nine patients with localized disease received chemotherapy with a combination of doxorubicin, methotrexate, cisplatin, and ifosfamide (eight adjuvant and one neoadjuvant chemotherapy) whereas four patients underwent radiation therapy (two adjuvant radiation 185 and two for palliation). One patient underwent selective arterial embolization with palliative intent.

The mean follow-up was 112 months (0–495), median 69 months. At the last follow-up, 14 patients out of 24 were dead of disease (DOD), 3 dead of other causes (DOC), one alive with metastatic disease (AWD), and 6 alive without evidence of disease (NED). Radiologically, all cases were lytic, with areas of sclerosis in two cases. Mean tumor size was 10.87 cm (range 5–20 cm); in 13 cases, it was ≤10cm while in the other 11 cases, it was >10 cm (Table 1).

### 3.2. Histopathological and Immunohistochemical Features

From a histopathological point of view, 15 cases showed more than 4 mitotic figures per 10 HPF and were associated with high cellularity, cytologic atypia, and >10% of necrosis, defining high-grade tumors (Figure 1, Table 2). CD34 and STAT6 immunopositivity was observed in 95% (23/24) and in 100% (24/24) of cases, respectively (Figure 2).

According to Demicco score [22], 8 patients (33%) were classified in the low-risk group, 11 (46%) in the intermediate-risk group, and 5 (21%) in the high-risk group (Table 1).

Two of the nine patients who developed distant metastasis belonged to the low-risk groups, five to the intermediate-risk group, while two patients belonged to the high-risk group. The three patients with metastasis at presentation were equally distributed in the three risk groups.

### 3.3. NAB2–STAT6 Fusion Variants

The analysis of fusion transcripts identified *NAB2–STAT6* fusion variants in 10 out of 12 (83.3%) samples (Table 2). In two cases, no variant was found. Considering the 24 types of fusion variants evaluated, 2 breakpoints were detected with a higher frequency: *NAB2exon6*—*STAT6exon17* (4 cases) and *NAB2exon4-STAT6exon2* (3 cases), followed by the breakpoint *NAB2exon6*—*STAT6exon16*, *NAB2exon2*—*STAT6exon2* and *NAB2exon6-STAT6exon16/NAB2exon6*—*STAT6exon17* in one case (Table 2). Regarding the Demicco score risk, the *NABex6-STAT6ex17* fusion variant was present only in high- and intermediate-risk patients, even if *NAB2-STAT6* fusion variants and Demicco score risk were not significantly correlated (*p* = 0.25).

### 3.4. TERT Promoter Mutations: C228T and C250T

The wild-type C250C genotype was shown in all 16 samples while no C250T mutations were detected. In only one DNA sample, a heterozygous C228T substitution was detected.

The only patient presenting this variant died one day after surgery due to complications; therefore, it was not possible to evaluate its prognostic role (Table 2).

### 3.5. p53 Mutations

Overall, we detected p53 genetic alterations in 11 samples (Table 2). Three samples presented point mutations: a nonsense heterozygous variant (p.Gln165*) was detected in patient 1; a missense heterozygous variants (p.Ala63Val), already described as a variant of uncertain significance (VUS), was detected in patient 16; and a homozygous splice site alteration (c.375 + 1G > A) was observed in patient 11. All samples except two (1 and 14) showed the presence of a copy number variation (CNV) involving at least exon 4 of *p53*. In detail, CNV deletions were detected in patient 2, 7, 13, 15, 18, 19, and 23 whereas CNV amplifications were detected in patient 16 and 21.

### 3.6. Correlations between Clinicopathological, Immunohistochemical, and Molecular Data

Regarding the entire population of study (24 cases), 5- and 10-year DSS were respectively 64% and 42%, whereas on the localized tumor, 5- and 10-year disease-related-specific DSS were respectively 80% and 60%. As expected, localized and surgically treated patients (16 out of 24, 66%) showed a better 5-year DSS than metastatic ones (74% vs. 33%) (Figure 3).

Table 3 summarizes the results of the Kaplan–Meier survival analysis of the clinicopathological variables (histological grade, tumor size, age, mitosis, necrosis, Demicco score risk). Stratification by tumor size did not correlate with DSS either for localized patients (*p* = 0.54) or for the whole series (*p* = 0.44). However, the only patient with tumor size <5 cm was alive at follow up (Table 3). Stratification based on mitotic count was carried out (A ≤ 1 mitosis, B = 1–3 mitosis, and C ≥ 4); no correlation was found in terms of DSS at the 5- and 10-year follow up either for the whole series (*p* = 0.54) or for patients with localized disease (*p* = 0.33) (Table 3).

No significant differences in terms of DSS were found between the different variables analyzed by univariate analysis. Of interest, DSS in patients aged <55 and ≥55 years was found to be almost near statistical significance (*p* = 0.06), confirming a better prognosis in younger patients. In line with the malignancy histological criteria, none of the *NAB2-STAT6* fusion variants detected were significantly correlated to DSS both in all 24 cases (*p* = 0.72) and in 16 localized cases (*p* = 0.57). In localized patients, Exon6 was involved in 2 cases out of 5 while other fusion variants (Exon2, Exon4, Other) were detectable in 3 cases out of 6; no significant correlation (*p* = 0.68) in terms of DSS was observed at the 5- and 10-year follow up (80% vs. 40% and 100% vs. 67%, respectively). *P53* variants were assessed in 12 cases: 11 (91.6%) cases showed variation while in one case, no alteration was found. Since few cases were analyzed, no statistical analysis was done; however, tumors with p53 mutations were classified as follows: two ‘low-risk’, three ‘high-risk’, and six ‘intermediate-risk’ cases. Further, 5- and 10-year DSS in the mutated patient was 73% and 54%, respectively, with a mean follow up of 139 months (range 8–495).

The MFS was found to be about 72% at 5 years and 27% at 10 years, as 9 out of 16 patients developed distant metastasis after a mean time of 53 months, whereas the RFS was found to be 100% at 5 years and 75% at 10 years, respectively, as 3 patients out of 16 developed local recurrence after a mean time of 106 months. No significant differences in terms of MFS and of RFS were found between the different variables analyzed by univariate analysis.

Of interest was finding that no local recurrence occurred in patients considered to be low-grade malignancy. In particular, 10-year RFS was 64% for high-grade patients against 100% in low-grade patients. Nevertheless, the *p* value obtained was not significant (*p* = 0.19), probably due to the limited number of patients, which could represent a bias.

## 4. Discussion

Primary B-SFT represents an extremely rare entity and to date any correlations between histopathological, immuno-histochemical, and molecular features and DSS have not yet been determined due to the lack of sufficiently numerous cases reported. Despite the rarity of this pathology, this is extremely important in order to stratify patients in terms of risk of relapse and distant metastasis and thus define the best treatment and surveillance strategies. To the best of our knowledge, the present study reports the clinical, histopathological, and molecular characteristics of the largest series reported in the literature. The primary aim of this work was to find any clinical and molecular prognostic factors and to compare them with those currently used for S-SFT [27], evaluating the possibility of a different behavior between SFT originating from the bone and from soft tissue, even if they share the same histology. Secondly, we applied the risk stratification system proposed by Demicco et al. in 2017 [22] to our selected series of 16 patients with resectable and localized primary B- SFT at onset, in order to understand whether this system, already evaluated by us previously on patients affected by S- SFT of the extremities, was able to correctly predict the risk of local and distant metastatic relapse even in the case B-SFT.

Despite the fact that SFTs of the bone and of soft tissue share the same morphological features, the data obtained in this series of the B-SFT did not confirm those already obtained by us on the S-SFT series, comparable to those available for other cases reported in the literature [14,15,18,22,24,29].

In particular, no correlations emerged between DSS, RFS, and MFS with clinicopathological variables (histological grade, tumor size, age, mitosis, necrosis, Demicco score risk), unlike what was reported by Gold and Barthelmess [14,25], and molecular features (*TERT* promoter mutations [14,21,25,29] and *NAB2-STAT6* fusion transcripts variants). These last results appear to be in line with those reported by Machado and Bianchi [21,27]. Data of interest was the absence of local recurrence in all low-risk patients (according to Demicco scoring system), although without evidence of statistical correlation.

*TERT* promoter mutation in a heterozygous state (C250C/C228T) was only found in one case out of 16. Unfortunately, the patient died the day after surgery due to complications, preventing assessment of the possibility of any correlations with this mutation. These data differ markedly from those obtained by Gold, Machado, and Barthelmess [14,21,25] and from those in the case series of S-SFT of the extremities presented by Bianchi and collaborators, in which the frequency of mutations of the promoter of *TERT* was found to be nearly 50% and 23.7%, respectively. In particular, all three metastatic patients of our previous study presented C228T site mutation in a homozygous state [29].

Considering p53, almost all evaluated samples (91.6%) showed a genetic variant, different from what has been reported in the literature [21]. Despite the limited number of samples, 9 of the 11 tumors with p53 mutations were classified as ‘high’ or ‘intermediate’, thus confirming results detailed in a previous study [21]. Further studies will be required to evaluate the inclusion of p53 genetic status in the risk stratification system. Regarding *NAB2-STAT6* fusion transcript variants, the most frequently encountered was *NAB2ex6-STAT6ex17* (4 out of 12 cases), followed by *NAB2ex4-STAT6ex2* in 3 cases, *NAB2ex2-STAT6ex2* in one case, *NAB2ex6-STAT6ex16/NAB2ex6-STAT6ex17* in one case, and *NAB2ex6-STAT6ex16* in another. In two cases, there was evidence of different breakpoints from the 24 most frequently evaluated, which were therefore encoded as the other. In contrast, in our S-SFT series, the most frequently reported variants were *NAB2ex6-STAT6ex17*, *NAB2ex6-STAT6ex2*, and *NAB2ex4-STAT6ex4* [27]. In both series, no statistically significant correlations emerged between the different fusion variants and the oncological outcome, different from Barthelmess and Tai, who reported a better prognosis for *NAB2ex4-STAT6ex2/4* variants associated with a lower mitotic count and relapse rate [25,29]. However, there was a tendency for the *NAB2ex6-STAT6ex17* fusion variants to be associated with high- and intermediate-risk neoplasms according to the Demicco system [22]; however, this did not result in statistically significant values (*p* = 0.25).

Among the limitations of this study, first, the population under study was small due to the extreme rarity of this pathology. Furthermore, there was a lack of uniformity regarding the type of treatment of the cases treated, because of the long period evaluated, during which the therapeutic approaches changed: most of the non-operated patients date back to the first decades of this period, while over the years, surgical treatments have gradually become more and more conservative and have been associated with adjuvant chemotherapy, and adjuvant or palliative radiation therapy in some cases. It is important to underline that ancillary genetic investigations, such as FISH and RT-PCR, are not yet of practical use in molecular diagnostics and are not always feasible in all cases because of the decalcification process that occurs on bone samples.

## 5. Conclusions

In conclusion, no correlation emerged between Demicco’s risk assessment criteria and clinical behavior as evidenced for the S-SFT. In fact, the clinicopathological criteria of malignancy devised for SFT of soft tissues failed to predict outcomes in primary SFT of the bone. Further validation on more numerous as well as more homogeneous samples is necessary to validate some molecular differences between primitive SFT of the bone with respect to that of soft tissues and to evaluate the eventual prognostic implications.

## Figures and Tables

**Figure 1 cancers-13-02470-f001:**
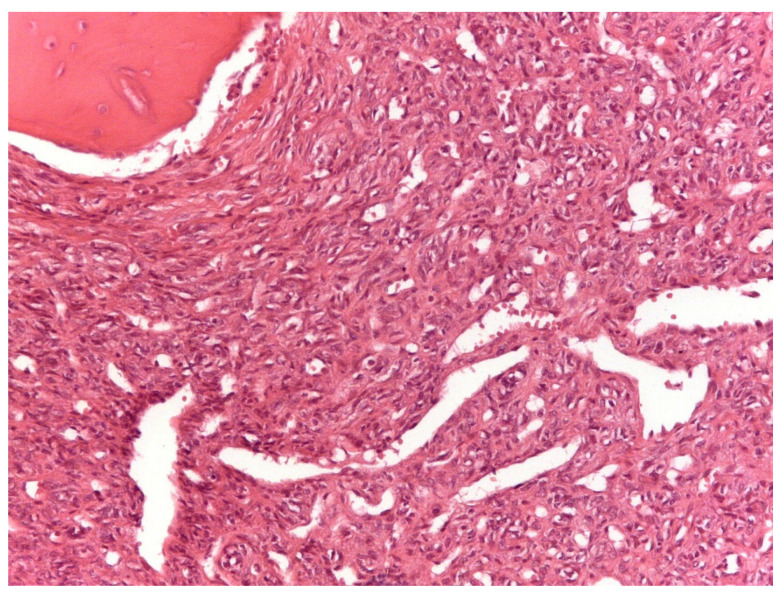
Solitary fibrous tumor: A spindle cell proliferation showing hemangiopericytoma-like blood vessels is seen (Hematoxylin &Eosin, original magnification, ×100).

**Figure 2 cancers-13-02470-f002:**
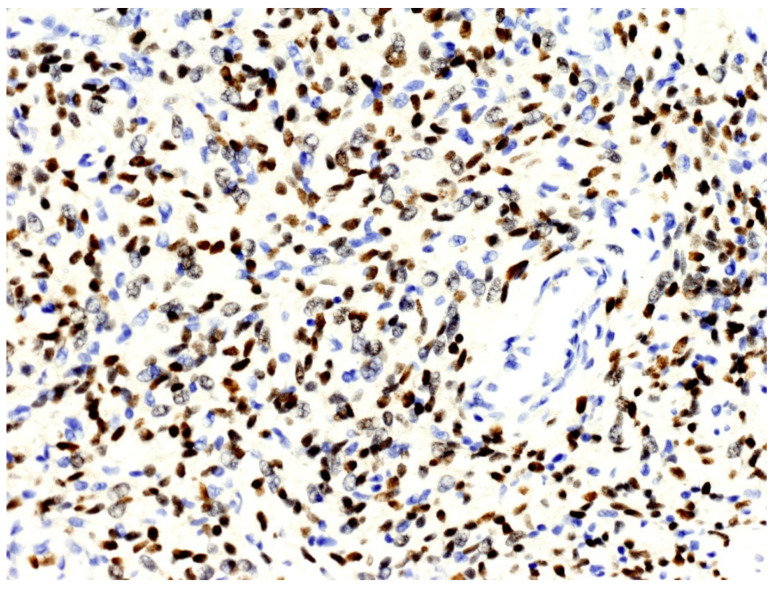
The nuclei of neoplastic cells express STAT6 (original magnification, ×200).

**Figure 3 cancers-13-02470-f003:**
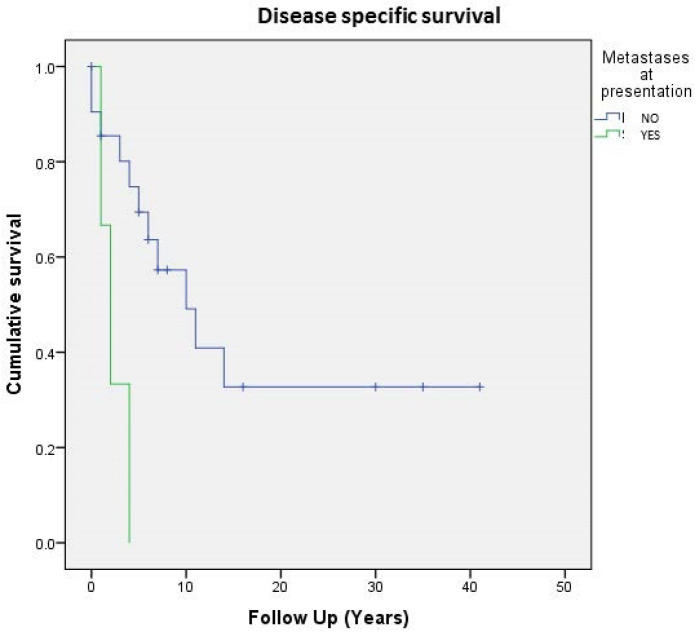
Cumulative survival of 24 patients affected by primary bone solitary fibrous tumor; stratification by metastases at presentation.

**Table 1 cancers-13-02470-t001:** Clinical features of 24 patients with a diagnosis of primary SFT of the bone.

Patient	Age	Sex	Anatomical Site	Tumor Size (cm)	Metastases	Surgical Procedure	Surgical Margins	DeMicco Score	Local Recurrence	Follow-Up (Months)	Status
1	75	F	Proximal tibia	15	Yes (3)	Thigh amputation	Wide	High	No	8	DOD
2	61	M	Distal femur	12.5	Yes (5)	Thigh amputation	Wide	High	No	15	DOD
3	66	M	Distal fibula	14	Yes (0)	Leg Amputation	Radical	Intermediate	No	58	DOD
4	71	F	Scapula	8	Yes (0)	Resection	Wide	Intermediate	No	28	DOD
5	84	M	Iliac wing	20	No	Inoperable	NA	Intermediate Inter	Yes (2)	2	DOD
6	52	F	Ileum pubic branch	7	No	Inoperable	NA	Low	Yes (24)	56	DOD
7	54	F	Femoral shaft	11	Yes (108)	Resection	Wide	Intermediate	Yes (149)	168	DOD
8	58	M	Sacrum	7.5	Yes (75)	Curettage	Marginal	Intermediate	Yes (97)	122	DOD
9	44	F	Distal femur	8.5	Yes (66)	Resection	Wide	Low	No	87	DOD
10	39	F	Proximal tibia	5	No	Resection	Wide	Low	No	421	NED
11	47	M	Scapula	6	No	Scapulectomy	Wide	Intermediate Intermediate	No	96	DOC
12	44	M	Sacrum	7	No	Inoperable	NA	Low	No	73	DOD
13	50	F	Proximal humerus	16	Yes (30)	ISTA	Radical	Intermediate	No	361	NED
14	58	M	Ischium pubic branch	8	No	Resection	Wide	Low	No	0	DOC
15	61	F	Iliac wing	8	Yes (84)	Resection	Intralesional	Low	No	137	DOD
16	27	F	Distal femur	17	No	IIAA	Intralesional	Intermediate	No	495	NED
17	32	F	Sacrum	15	No	Inoperable	NA	High	No	41	DOD
18	54	M	Sacrum	10	No	Resection	Wide	Low	No	67	NED
19	49	M	Proximal humerus	19	No	ISTA	Radical	High	No	92	DOC
20	31	F	4th lumbar vertebra	8	Yes (38)	Vertebrectomy	Wide	Low	No	66	DOD
21	21	F	Proximal humerus	9.5	Yes (72)	Resection	Wide	Intermediate	Yes (72)	72	AWD
22	7	F	Humeral shaft	11	No	Inoperable	NA	Intermediate	No	196	NED
23	60	F	Proximal femur	7.3	Yes (0)	Resection	Wide	Intermediate	No	21	DOD
24	69	M	Femoral shaft	10.5	No	Resection	Wide	High	No	18	NED

Legend: F: female; M: male; ISTA, interscapulothoracic amputation; IIAA, interileoabdominal amputation; NA, not applicable; AWD, Alive with disease; NED, non-evidence of disease; DOD, dead of disease; DOC, dead of other causes.

**Table 2 cancers-13-02470-t002:** Histopathological and molecular features of SFT patients.

Patient	Mitotic Count (X10 HPF)	Necrosis (≥10%)	STAT6	CD34	*TERT* Mutation	*NAB2-STAT6* Fusion	P53 Variants (Variant Type)	Grade of Malignancy
1	≥4	Yes	Pos	Pos	C250C/C228C	NA	yes (nonsense)	HIGH
2	≥4	Yes	Pos	Pos	C250C/C228C	EX6-EX17	yes (CNV deletion)	HIGH
3	<4	Yes	Pos	Pos	NA	NA	NA	LOW
4	≥4	Yes	Pos	Pos	C250C/C228C	NA	NA	HIGH
5	<4	No	Pos	Pos	NA	NA	NA	LOW
6	<4	No	Pos	Pos	NA	NA	NA	LOW
7	≥4	Yes	Pos	Pos	C250C/C228C	EX2-EX2	yes (CNV deletion)	HIGH
8	≥4	Yes	Pos	Pos	C250C/C228C	EX6-EX16	NA	HIGH
9	<4	No	Pos	Neg	NA	NA	NA	LOW
10	≥4	No	Pos	Pos	NA	NA	NA	HIGH
11	≥4	Yes	Pos	Pos	C250C/C228C	NA	yes (splice site)	HIGH
12	<4	No	Pos	Pos	C250C/C228C	EX4-EX2	NA	LOW
13	≥4	No	Pos	Pos	C250C/C228C	NA	yes (CNV deletion)	HIGH
14	<4	No	Pos	Pos	C250C/C228T	OTHER	no	LOW
15	<4	No	Pos	Pos	C250C/C228C	OTHER	yes (CNV deletion)	LOW
16	<4	Yes	Pos	Pos	C250C/C228C	EX6-EX16/EX6-EX17	yes (missense + CNV amplification)	LOW
17	≥4	Yes	Pos	Pos	NA	NA	NA	HIGH
18	<4	No	Pos	Pos	C250C/C228C	EX4-EX2	yes (CNV deletion)	LOW
19	≥4	Yes	Pos	Pos	C250C/C228C	EX6-EX17	yes (CNV deletion)	HIGH
20	≥4	No	Pos	Pos	NA	NA	NA	HIGH
21	≥4	Yes	Pos	Pos	C250C/C228C	EX6-EX17	yes (CNV duplication)	HIGH
22	≥4	No	Pos	Pos	NA	NA	NA	HIGH
23	≥4	Yes	Pos	Pos	C250C/C228C	EX6-EX17	yes (CNV deletion)	HIGH
24	≥4	Yes	Pos	Pos	C250C/C228C	EX4-EX2	NA	HIGH

Legend: HPF: high power fields; Pos, positive; Neg, negative; NA, not applicable; MAL, malignant.

**Table 3 cancers-13-02470-t003:** Disease-specific survival (DSS) analysis related to clinicopathological parameters.

Variables		Disease Specific Survival (24 pts)			Localized Disease * (16 pts)		
		5 Years-DSS	10 Years-DSS	*p*-Value	5 Years-DSS	10 Years-DSS	*p*-Value
Histological Grade							
	Low	62%	31%	0.52	100%	67%	0.84
	High	65%	58%		82%	71%	
Size							
	(A) 0–4.99 cm	100%	100%	0.44	100%	100%	0.54
	(B) 5–9.99 cm	70%	36%		87%	45%	
	(C)10–14.99 cm	62%	62%		80%	80%	
	(D) >15	50%	50%		50%	50%	
Age							
	<55 years	86%	27%	0.06	100%	77%	0.15
	≥55 years	61%	27%		100%	60%	
Mitosis							
	(A) <1	60%	30%	0.54	60%	30%	0.33
	(B) 1–3	66%	33%		100%	50%	
	(C) ≥4	65%	58%		76%	68%	
Necrosis							
	<10%	80%	47%	0.66	100%	62%	0.95
	≥10%	51%	51%		78%	78%	
Gene Fusion							
	Exon6				80%	40%	0.68
	Other				100%	67%	
Demicco Score Risk							
	High				54%	54%	0.43
	Intermediate				72%	46%	
	Low				64%	28%	

* Surgically treated patients; DSS: disease-specific survival.

## Data Availability

Data available on request due to restrictions e.g., privacy or ethical. The data presented in this study are available on request from the corresponding author.

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
