# Peer review of "Clinical, Histological, and Molecular Features of Solitary Fibrous Tumor of Bone: A Single Institution Retrospective Review"

_cancers, 2021, doi:10.3390/cancers13102470_

Round 1

Reviewer 1 Report

The response and changes made as per comments are acceptable. 

Author Response

The response and changes made as per comments are acceptable.

I would like to thank the reviewer for this comment.

Reviewer 2 Report

There are some comments.

The number of the expression of CD34 and STAT6 do not match between text and Table 2. Please check.

It would be better to evaluate further based on mitotic counts number 0, 1-3, and ≥ 4/10 HPF as Dr. Demicco et al. analyzed.

Please correct NAB2-STAT6 and TERT gene in italics.

Author Response

REVIEWER 2

There are some comments.

The number of the expression of CD34 and STAT6 do not match between text and Table 2. Please check.

Following the Reviewer's suggestion, we checked this point and we corrected the results in abstract and in "results" section, "Histopathological and immunohistochemical features" paragraph line 211.

It would be better to evaluate further based on mitotic counts number 0, 1-3, and ≥ 4/10 HPF as Dr. Demicco et al. analyzed.

We thank the Reviewer for his/her comment. We have made a mistake in text and in table 3, because we evaluated the mitotic counts number according Demicco criteria (0,1-3 and ≥ 4/10 HPF) but we wrote in text and in table 3 in a wrong way reporting 1, 2-3 and ≥ 4/10 HPF. For this, we corrected the text (results section, "Correlations between clinicopathological, immunohistochemical, and molecular data" paragraph, line 261) and table 3.

Please correct NAB2-STAT6 and TERT gene in italics.

Following the Reviewer's suggestion, we modified it.

Reviewer 3 Report

The authors addressed all the concerns. However, they did not include the new p53 analyses in the abstract.

Author Response

REVIEWER 3

The authors addressed all the concerns. However, they did not include the new p53 analyses in the abstract.

I would like to thank the reviewer for this comment and following his/her suggestion we added p53 analyses in the abstract.

This manuscript is a resubmission of an earlier submission. The following is a list of the peer review reports and author responses from that submission.

Round 1

Reviewer 1 Report

In this manuscript the authors have characterized 24 cases of primary bone SFT.  

Multiple spelling/english errors throughout the manuscript need to be corrected as below. 

1) HPC/TFS needs to be expanded in line 71. It has not been used earlier in the manuscript 

2) Line 169, 270, 271,272,275,276 TFS needs to be corrected

3) Tumor size in table 1 has comma instead of decimals.  

4) What kind of chemotherapy was given (line 181)?

5) Language needs to be revised in 184, 319, 320

6) In line 152 if patients who died of other causes were excluded then the correct statistic measure should be disease specific survival rather than overall survival  

Author Response

Response to Reviewer 1 Comments

In this manuscript the authors have characterized 24 cases of primary bone SFT.  

Multiple spelling/english errors throughout the manuscript need to be corrected as below. 

1) HPC/TFS needs to be expanded in line 71. It has not been used earlier in the manuscript.

Following the Reviewer's suggestion, we reviewed this point in line 71.

2) Line 169, 270, 271,272,275,276 TFS needs to be corrected.

Following the Reviewer's suggestion, we corrected this abbreviation in these lines.

3) Tumor size in table 1 has comma instead of decimals.  

Following the Reviewer's suggestion, we modified tumor size in table 1.

4) What kind of chemotherapy was given (line 181)?

The patients underwent to "osteosarcoma-like" protocol of chemotherapy with a combination of doxorubicin, methotrexate, cisplatin and ifosfamide and we added this data in "3.1 paragraph: Clinicopathological evaluation" (lines 190-191 and 194-197).

5) Language needs to be revised in 184, 319, 320.

Following the Reviewer's suggestion, we revised lines 184, and 319, 320.

6) In line 152 if patients who died of other causes were excluded then the correct statistic measure should be disease specific survival rather than overall survival.

We thank the Reviewer for his/her comment, according to which we have clarified this point using " disease specific survival ".

Reviewer 2 Report

The manuscript showed an interesting study about clinicopathological features of solitary fibrous tumor (SFT) of primary bone and was well written.

For comparison with soft tissue SFT, it would be better to analyze the survival correlation between risk stratification on the base of mitotic count (1, 1-3,  ≥4/ 10 HPFs) and tumor size (0–4.9, 5–9.9, 10–14.9, ≥ 15 cm) of Demicco's risk assessment criteria. 

Minor points:
 Please explain the abbreviation for HPC/TFS.
 Please check English grammar 
       e.g., 5 femur 3 tibia and 1 fibula -> 5 femur, 3 tibia, and 1 fibula

Author Response

The manuscript showed an interesting study about clinicopathological features of solitary fibrous tumor (SFT) of primary bone and was well written.

For comparison with soft tissue SFT, it would be better to analyze the survival correlation between risk stratification on the base of mitotic count (1, 1-3,  ≥4/ 10 HPFs) and tumor size (0–4.9, 5–9.9, 10–14.9, ≥ 15 cm) of Demicco's risk assessment criteria. 

Following the Reviewer's suggestion, we have done these correlations and the relative results has been included in paragraph "3.6. Correlations between clinicopathological, immunohistochemical, and molecular data" (lines 279-284) and in Table 3.

Minor points:
Please explain the abbreviation for HPC/TFS.

Following the Reviewer's suggestion, we explained this abbreviation.

Please check English grammar 
       e.g., 5 femur 3 tibia and 1 fibula -> 5 femur, 3 tibia, and 1 fibula

Following the Reviewer's suggestion, we explained English grammar.

Reviewer 3 Report

This study describes clinical, pathological and molecular findings in a series of SFT arising in bone, which do not show clear correlations with tumor behavior in contrast to what is reported for larger series of soft tissue SFT. The major limitation of the study is the small number of patients, owing to the extreme rarity of this histological subtype of SFT, thus precluding the possibility of accurate patient risk stratification. The tumor series could have been expanded by adding cases from other institutions. However, this drawback could be counterbalanced by assessing additional molecular markers that has been previously reported to impact SFT patient prognosis. As the authors indicate in the Introduction section TERT promoter mutations could have also prognostic value. Indeed, it has been proposed that risk stratification systems could be improved by incorporating molecular data such us TERT promoter mutations or p53 mutations to the current models, especially to refine stratification of patients within intermediate-risk groups. Since TERT mutations were only detected in a unique patient, molecular information should be complemented with p53 mutations which may be relevant and therefore added to the risk analyses. Other molecular markers such as APAF1 could be considered as well. Moreover, other risk stratification models available should be tested in order to ascertain which one is the best candidate in this context to be improved with additional molecular information.

Regarding gene fusion, it may be more accurate grouping different exonic variants by considering lost/retained functional protein domains, thus unifying different exonic variants with similar protein domain composition into fewer groups to increase sample size for statistics. Score risk for fusion variants should be also tabulated for clarity. As sample size is limiting, it would be worth considering identifying the gene fusion by targeted RNAseq in RT-PCR negative cases, just in case those cases express fusion variants not covered by the primer pairs used.

Minor points:

  • Clinical information in section 3.1 is a bit confusing. It should be indicated in the table whether patients presented localized disease, developed metastasis, recurrence, etc, in order to make more understandable subsequent analyses.
  • Authors indicate the histological criteria to label a case as dedifferentiated SFT (line 95-98). However, there is no mention in the manuscript to specific associations of cases with this feature and other parameters in the study.
  • A bibliographic reference is missing in the section 2. Immunohistochemistry (line 101).
  • Table 2: patient #16 shows two fusion variants, but the second one is not completely specified.

Author Response

Response to Reviewer 3 Comments

This study describes clinical, pathological and molecular findings in a series of SFT arising in bone, which do not show clear correlations with tumor behavior in contrast to what is reported for larger series of soft tissue SFT. The major limitation of the study is the small number of patients, owing to the extreme rarity of this histological subtype of SFT, thus precluding the possibility of accurate patient risk stratification. The tumor series could have been expanded by adding cases from other institutions. However, this drawback could be counterbalanced by assessing additional molecular markers that has been previously reported to impact SFT patient prognosis. As the authors indicate in the Introduction section TERT promoter mutations could have also prognostic value. Indeed, it has been proposed that risk stratification systems could be improved by incorporating molecular data such us TERT promoter mutations or p53 mutations to the current models, especially to refine stratification of patients within intermediate-risk groups. Since TERT mutations were only detected in a unique patient, molecular information should be complemented with p53 mutations which may be relevant and therefore added to the risk analyses. Other molecular markers such as APAF1 could be considered as well. Moreover, other risk stratification models available should be tested in order to ascertain which one is the best candidate in this context to be improved with additional molecular information.

 As suggested by the reviewer, we included in the study a molecular characterization of p53 gene performed by Sanger sequencing and Digital PCR technique, due to the availability of an already set up method. All informations about the protocol have been added in the ‘Materials and Methods’ section, as well as a brief description and analysis of results in ‘Results’ and ‘Discussion’ section. We have also implemented the Table 2 including also p53 results.

Regarding gene fusion, it may be more accurate grouping different exonic variants by considering lost/retained functional protein domains, thus unifying different exonic variants with similar protein domain composition into fewer groups to increase sample size for statistics. Score risk for fusion variants should be also tabulated for clarity. As sample size is limiting, it would be worth considering identifying the gene fusion by targeted RNAseq in RT-PCR negative cases, just in case those cases express fusion variants not covered by the primer pairs used.

As suggested by the reviewer, RNAseq analysis it would be the best choice even if , due to the limited sample size,  the information would be purely descriptive, but RNAseq analysis was not feasible because of the lack of sufficient material.

On the base of reviewer’s suggestions, we have analyzed the EX-6 –xxx fusion variants versus other variants and we have reported in the paragraph "3.6. Correlations between clinicopathological, immunohistochemical, and molecular data" (lines 290-293) and in Table 3.

Minor points:

  • Clinical information in section 3.1 is a bit confusing. It should be indicated in the table whether patients presented localized disease, developed metastasis, recurrence, etc, in order to make more understandable subsequent analyses.

Following the Reviewer's suggestion, we revised table 1 indicating patients that presented localized disease, developed metastasis and recurrence.

  • Authors indicate the histological criteria to label a case as dedifferentiated SFT (line 95-98). However, there is no mention in the manuscript to specific associations of cases with this feature and other parameters in the study.

We thank the Reviewer for his/her comment, according to which we have removed this sentence to the histological criteria.

  • A bibliographic reference is missing in the section 2. Immunohistochemistry (line 101). Following the Reviewer's suggestion, we added a bibliographic reference.
  • Table 2: patient #16 shows two fusion variants, but the second one is not completely specified.

We thank the Reviewer for his/her comment, according to which we have specified this data in table 2.